# Decoding Diffusion: A Scalable Framework for Unsupervised Analysis of Latent Space Biases and Representations Using Natural Language Prompts

**E. Zhixuan Zeng, Yuhao Chen & Alexander Wong**
University of Waterloo
{ezzeng, yuhao.chen1, alexander.wong}@uwaterloo.ca

## Abstract

Recent advances in image generation have made diffusion models powerful tools for creating high-quality images. However, their iterative denoising process makes understanding and interpreting their semantic latent spaces more challenging than other generative models, such as GANs. Recent methods have attempted to address this issue by identifying semantically meaningful directions within the latent space. However, they often need manual interpretation or are limited in the number of vectors that can be trained, restricting their scope and utility. This paper proposes a novel framework for unsupervised exploration of diffusion latent spaces. We directly leverage natural language prompts and image captions to map latent directions. This method allows for the automatic understanding of hidden features and supports a broader range of analysis without the need to train specific vectors. Our method provides a more scalable and interpretable understanding of the semantic knowledge encoded within diffusion models, facilitating comprehensive analysis of latent biases and the nuanced representations these models learn. Experimental results show that our framework can uncover hidden patterns and associations in various domains, offering new insights into the interpretability of diffusion model latent spaces.

## 1 Introduction

Recent breakthroughs in image generation have significantly advanced the field of computer vision. In particular, Latent Diffusion Models (LDMs) [1] have emerged as powerful tools for generating high-quality and diverse images. Their success has led to widespread adoption across various creative and practical applications, from digital art and design to advertising and content creation. However, the proliferation of AI-generated images has also raised concerns about biases embedded within these models, as they often mirror societal stereotypes and reinforce existing prejudices when generating representations of people, professions, or other sensitive concepts. These biases are especially problematic because diffusion models are increasingly used in contexts where image outputs may influence public perception, media representation, and even automated decision-making processes. Thus, understanding the biases and limitations inherent in these generative models is crucial to ensuring ethical and responsible use.

Compared to Generative Adversarial Networks (GANs) [2], with a wealth of works analyzing and utilizing its latent spaces [3–8], diffusion models pose unique challenges. The iterative denoising process in diffusion models makes their latent space more complex and less interpretable, posing difficulties in understanding and analyzing the knowledge encoded within these models. This complexity hinders efforts to grasp how these models represent and manipulate concepts throughout the generation process [9]. This lack of transparency hinders efforts to fully grasp how diffusion models represent and manipulate concepts in their latent spaces. Recent research has addressed these

Preprint. Under review.

challenges by exploring latent directions in diffusion models to identify semantically meaningful features [10, 11]. However, these methods have several limitations that restrict their utility and insight:

1. **Ambiguous Directions:** Some approaches, such as Park *et al.*[11] and NoiseCLR [10], identify a set of potentially semantically important directions within the latent space. While these directions can reveal meaningful concepts, they often require extensive manual interpretation to understand their significance. Moreover, if the output images are unclear or ambiguous, the process must be repeated iteratively. As such, extracting actionable insights becomes a tedious and highly manual task.

2. **Limited Number of Vectors:** Other methods take a more explicit approach by training specific vector directions for known concepts, such as gender [12] or style changes [13]. While these approaches offer more precise control over image editing, they are limited by the small number of predefined vectors they rely on, often only representing one or two concept directions. Each new latent direction requires further training. This narrow focus makes them suitable for image manipulation tasks but less effective for broader exploration or understanding of the model's biases and knowledge representation.

This paper proposes a new framework for unsupervised exploration of diffusion model semantic latent spaces that directly ties latent directions to natural language prompts or image captions. Unlike previous methods that require manual interpretation or predefined vector training, our approach enables automatic and interpretable analysis of latent spaces by leveraging semantic information in language. This framework provides two key advantages:

- **Automatic Interpretation:** By directly associating latent directions with natural language, we eliminate the need for manual inspection and interpretation of each latent direction, making it easier to extract actionable insights.

- **Breadth of Exploration:** Our method can capture various latent directions corresponding to various concepts without requiring explicit training, enabling a more comprehensive understanding of how diffusion models encode semantic information across diverse contexts.

Through systematic experiments and evaluations, we demonstrate that our approach offers a more scalable and interpretable method for understanding diffusion latent spaces, paving the way for more transparent and robust generative models.

## 2 Related work

### 2.1 Semantic Directions

Semantic directions involve identifying vectors on some latent space corresponding to a concept, where adding or subtracting that vector would result in meaningful semantic change. This idea has existed for many years, with the famous example being "King - male + female = Queen" [14]. Word2Vec [15, 16] is an early pioneer of this idea and has shown its success in natural language relations. This is also a popular method in GANs [3]. Semantic latent directions have been used for face editing [8], shift camera angles and scales [5, 6], analyze memorability [7] and interpolate between images [4, 17, 18].

### 2.2 Semantic Directions in Diffusion Models

Applying the concept of semantic directions to diffusion models have been more challenging. SEGA [19] and Concept Algebra [13] identify semantic latent vectors for target concepts. However, these are limited in the number of vectors that can be trained, and are more suited for guiding image generation than exploring knowledge representation. NoiseCLR [10] introduces a way to identify semantically meaningful directions given a small dataset of images through contrastive learning. However, each identified direction requires manual interpretation to describe what it represents.

Kwon *et al.*[9] introduced the H-space, which is the output of the middle layer of the U-Net in the diffusion model. This space is information-rich and also has important properties including homogeneity, linearity, robustness, and consistency across timesteps. Subsequent papers have used

this space to discover semantic directions [11], condition the image output [12, 20], and more [21]. These methods either also require the same manual interpretation [11] as in NoiseCLR [10], or again only train for a few concepts, and are not scalable or flexible enough to perform large-scale analysis of the model's knowledge representation.

# 3  Methodology

The h-space is introduced by Kwon *et al.*[9] and is defined as the output of the middle U-Net layer in the diffusion model. This space represents a condensed, information-rich portion of the model's latent space, capturing intermediate representations during denoising. It also has nice properties like homogeneity, linearity, robustness, and consistency across timesteps. By sampling from this space, we aim to examine the biases and knowledge encoded within the diffusion model.

To sample from the h-space, we condition the model on a natural language prompt. At a given timestep, we then record the output of the middle U-Net layer, allowing us to observe the evolution of the h-space throughout the diffusion process.

However, beyond the prompt, two key factors influence the h-space vectors: (1) the timesteps in the diffusion process, and (2) the random seed. Since our goal is to focus on the effects of the prompt, we aim to minimize the impact of both timesteps and random seed variability on our analysis.

## 3.1  Timesteps

The diffusion process is iterative, with each timestep progressively denoising the input. Early timesteps contain more noise, while later steps refine the image's details. As such, each timestep may focus on different aspects of the image and can be unpredictable and noisy. And while a carefully trained h-space vector for a particular concept can remain consistent across timesteps for image editing purposes [9], directly sampling from the h-space remains noisy and inconsistent. This makes extracting semantic information from the h-space vectors more difficult.

To address this, we apply use **Latent Consistency Models (LCMs)** [22] through the latent consistency model LORA [23] to stabilize the h-space representations. Latent Consistency Models directly predict the solution of the Probability Flow ODE (PF-ODE) at $t = 0$, based on a consistency function $f_\theta(z, c, t)$ that maps from a noisy sample at any timestep $t$ to its denoised state. This approach helps reduce variability across timesteps and ensures that h-space samples reflect meaningful semantic information rather than noise-driven distortions.

To improve image quality, LCM samplers add noise back into the image after the first prediction and denoise again to achieve a final image after 2-6 iterations. For sampling the h-space vectors when using LCM, only the first timestep is saved.

Since Latent Consistency models requires fewer timesteps to generate an image, it is also more efficient for sampling.

## 3.2  Seed

The random seed determines the initialization of noise added during the forward diffusion process, introducing stochastic variations that affect both the generated images and the intermediate h-space representations. To focus solely on the prompt's influence, we control for the random seed by only comparing vectors from the same seed. That difference vector can then be averaged across multiple seeds to mitigate the impact of individual noise configurations and to reveal consistent patterns in how the prompt modifies the h-space. This process allows us to isolate prompt-driven variations, providing a more robust understanding of how the model encodes semantic concepts and biases in its latent representations.

By minimizing the impact of the random seed and timestep variability, we ensure that our analysis of the h-space centers on the prompt's influence, providing a clearer understanding of the model's biases and the encoded knowledge.

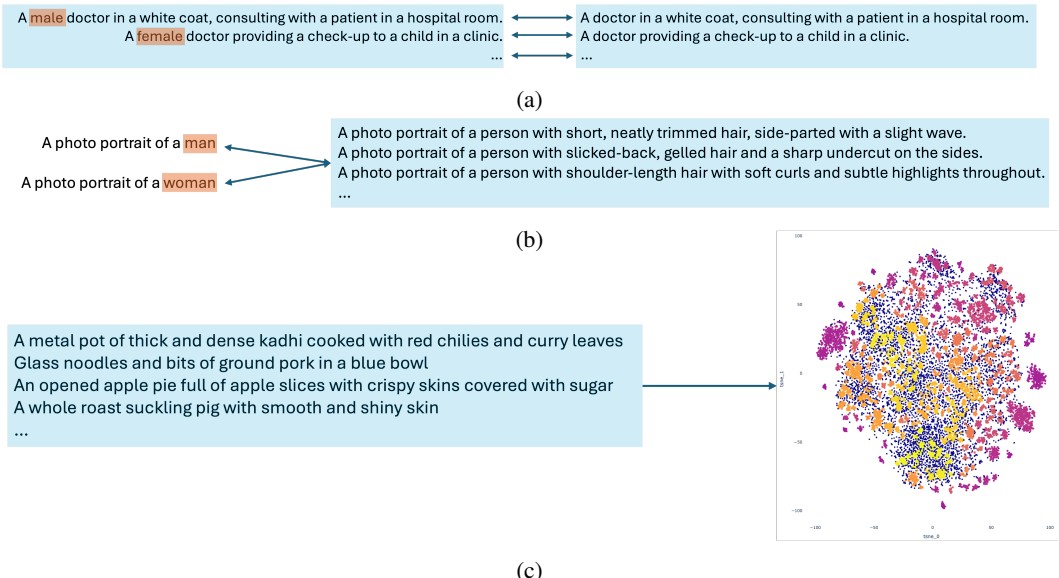

Figure 1: Overview of the three experimental approaches for analyzing h-space representations. (a) One-to-one comparisons isolate the impact of adding or removing a single concept. (b) One-to-many comparisons rank multiple prompts along a semantic spectrum. (c) Clustering captures broader patterns and associations across diverse captions.

# 4 Experiments/Analysis

After obtaining the h-space vectors, this section presents several analyses that can be performed to gain deeper insights into the biases and knowledge encoded by the diffusion model. As illustrated in Figure 1, we explore three distinct approaches for analyzing the latent space: (a) one-to-one comparisons between prompts with and without specific concepts, (b) one-to-many comparisons to assess how a set of descriptions aligns with key semantic concepts, and (c) cluster visualizations to reveal naturally occurring groupings of captions and their shared attributes.

## 4.1 One To One Comparison

One approach to analyze the h-space vectors is through one-to-one comparisons between prompts that either include or exclude a specific concept. In this experiment, descriptions of individuals in various professions are generated for both male and female subjects using ChatGPT-4 [24]. H-space vectors are then sampled for these prompts. Next, words explicitly referencing gender are removed, and the h-space vectors are sampled again for these gender-neutral descriptions.

To quantify the influence of gendered terms, we compute the cosine distance between the h-space vectors corresponding to the gendered and gender-neutral prompts. This distance highlights the extent of gender bias present in the model's representations for different professions, revealing implicit associations even when gender is not explicitly stated in the text. Figure 2 illustrates these biases across a range of professions, demonstrating how the model's internal representations differ based on subtle prompt variations.

We validate these biases by classifying the images generated from non-gendered prompts using CLIP [25]. Professions with the largest differences between male and female cosine distances (favoring male representations) in Figure 2 show the lowest probabilities of generating female-presenting images (see Table 1). This indicates that greater divergence in h-space representations correlates with a decreased likelihood of producing female images when given gender-neutral prompts.

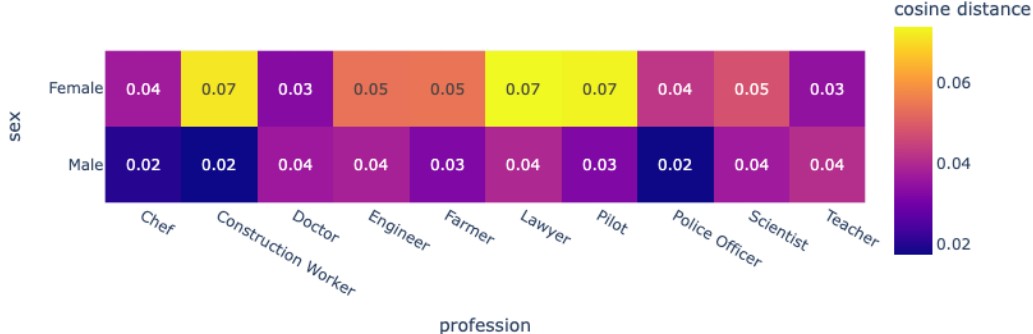

Figure 2: Average cosine distance between h-space vectors sampled from prompts with and without explicit gender references, grouped by gender and profession. Female representations for several professions (e.g., lawyer, pilot, construction worker) show greater deviation from the non-gendered "default" representation compared to male counterparts, indicating stronger biases favoring male stereotypes in these roles.

| Profession | % female images | Cosine Distance (Female - Male) |
|---|---|---|
| Teacher | 58.33% | -0.0003 |
| Scientist | 41.67% | 0.0088 |
| Chef | 11.67% | 0.0138 |
| Lawyer | 25.00% | 0.0145 |
| Farmer | 23.33% | 0.0153 |
| Doctor | 45.00% | 0.0170 |
| Police Officer | 10.00% | 0.0295 |
| Engineer | 1.67% | 0.0394 |
| Construction Worker | 1.67% | 0.0428 |
| Pilot | 1.67% | 0.0470 |

Table 1: The difference in cosine distances between female and male representations from Figure 2 strongly correlates with the percentage of female-presenting images generated from non-gendered prompts. A smaller cosine distance is associated with a higher probability of generating a female image, while a larger distance corresponds to a much lower probability.

## 4.2 One to many comparison

Sometimes, directly isolating and removing a specific concept from every prompt may be challenging. Instead, we can compare a given concept against a set of diverse prompts.

In the following example, ChatGPT 4o [24] was used to generate gender-neutral descriptions of facial portraits. H-space vectors were then collected using these captions. Those results were then compared with h-space vectors generated for each gender, namely: "a photo portrait of a man" and "a photo portrait of a woman". The difference in cosine similarity between each gender-neutral prompt and the two gendered prompts was then used to rank the gender-neutral prompts from "most similar to man" to "most similar to woman" (Table 2).

The use of cosine distance differences, rather than directly comparing the distance to male or female vectors independently, is necessary because some prompts may be equally distant from both gender vectors for reasons unrelated to gender. For example, if "a photo portrait of a man" and "a photo portrait of a woman" both produce black-and-white images, but a given prompt results in a colored image, the colored image will be far from both male and female vectors in h-space. By using the difference between male and female distances, we minimize the impact of these unrelated factors.

When asked to characterize the differences between two hypothetical people based only on the distances to each caption, ChatGPT-4 provided the following summary:

| caption | Cosine Difference (Female - Male) |
|---|---|
| A photo portrait of a person with a bald head and a light sheen, complemented by a well-groomed beard. | 0.20140 |
| A photo portrait of a person with slicked-back, gelled hair and a sharp undercut on the sides. | 0.11873 |
| ... | ... |
| A photo portrait of a person with shoulder-length hair in loose waves, casually tucked behind the ears. | -0.18553 |
| A photo portrait of a person with shoulder-length hair with soft curls and subtle highlights throughout. | -0.20819 |

Table 2: A number of captions describing hair, ranked from closest to male to closest to female

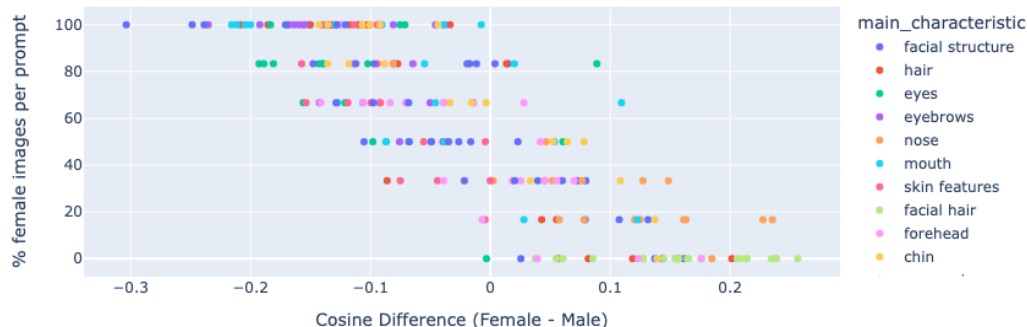

Figure 3: Correlation between the percentage of images classified as female (using CLIP) and the difference in cosine distances between gendered h-space vectors for each prompt. Prompts with higher cosine differences favouring female vectors are more likely to generate female-presenting images, indicating a strong association between h-space distances and perceived gender.

> Person A is more likely to have hairstyles associated with a cleaner, sharper, or minimalist look. Specifically, bald or buzz cut, slicked-back or neatly styled hair, and shorter, simpler haircuts. Person B is more likely to have more complex or varied hairstyles, particularly longer and styled in different ways. Characteristics associated with person B include longer hair, more intricate styling, textured and curly hair, and more highlights or coloring.

Similar to the one-to-one comparisons in Section 4.1, we also validate these findings using CLIP [25] for gender classification. Figure 3 plots the percentage of images classified as female against the difference in cosine distances of the h-space vectors, showing a strong correlation between the two metrics.

## 4.3 Cluster Visualization

The above two use cases both involve targeted analysis of bias for a particular concept through specially crafted captions. However, we can also analyze more naturally occurring captions through visualizing the clusters that may be present.

The following section uses the Food500-CAP [26] dataset, which contains natural language captions for 24700 food image. After sampling the h-space vectors for a given seed, we can use a visualization tool such as t-SNE [27] to visualize the high-dimension data in 2D space. Clustering algorithms such as HDBSCAN [28–30] can be used to automatically group points to form clusters.

Figure 4 demonstrates this visualization and points out several prominent clusters. Because each cluster is made up of captions instead of images, we can easily identify the common element in a

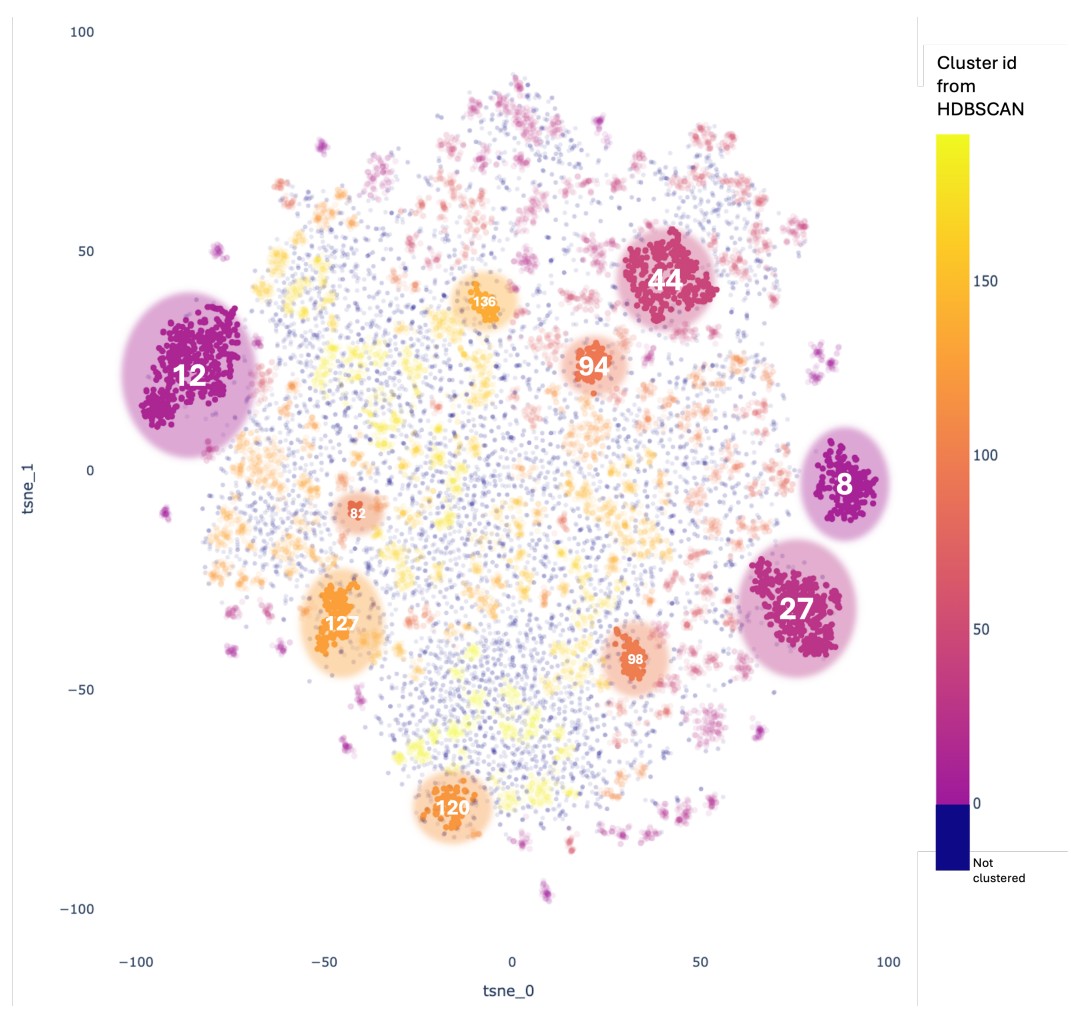

Figure 4: t-SNE visualization of h-space vectors sampled from the Food500-CAP dataset. Clusters are labelled based on the id assigned by HDBSCAN [30]. Some prominent clusters include (8) square dishes; (12) pots with soup; (27) sandwiches/bread; (44) pie; (94) takeout boxes; (98) rectangular plates; (120) salads; (127) stir fried noodles; and (144) seafood boil; (136) purple vegetables, esp. purple cabbage

cluster through LLMs such as ChatGPT [24]. This removes the need to manually "describe" the latent vectors through observation, like Kwon *et al.*[9] and NoiseCLR [10].

For each cluster, it is also possible to take the average H-space vector of that cluster and add that back into the model to condition the output. The result of this can be seen in Figure 5. It is also possible to combine different clusters to get a combined result, as seen in Figure 5e.

This experiment demonstrates that even without explicitly targeting a specific concept, naturally occurring captions can reveal latent biases and associations. By leveraging clustering and visualization techniques, we can map out the conceptual landscape of the h-space, offering a broader understanding of how diffusion models encode and express complex ideas.

## 5    Conclusion

In this paper, we introduced a new framework for unsupervised exploration of diffusion model latent spaces using natural language prompts and image captions. Unlike previous methods that require manual interpretation or rely on a limited number of predefined vectors, our approach enables automatic and interpretable analysis of latent directions by directly associating them with semantic

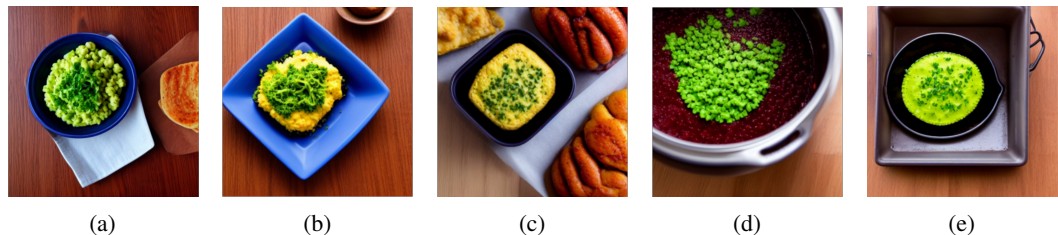

| (a) | (b) | (c) | (d) | (e) |

Figure 5: (a) default image for "a photo of food" with seed 0, (b) image conditioned on "a photo of food" plus the average of cluster 8 from Figure 4, depicting a square plate (c) image conditioned on "a photo of food" plus the average of cluster 27 from Figure 4, depicting bread (d) image conditioned on "a photo of food" plus the average of cluster 12 from Figure 4, depicting a pots (e) image conditioned on "a photo of food" plus the average of clusters 8 and 12 from Figure 4, depicting a square pot

information derived from language. This framework not only simplifies the process of understanding latent representations but also expands the range of concepts that can be explored, offering a more comprehensive view of the knowledge encoded within diffusion models.

Through various experiments, we demonstrated the effectiveness of our approach in identifying biases, understanding complex associations, and visualizing semantic clusters within diffusion latent spaces. The ability to systematically map these directions to interpretable concepts opens new avenues for understanding and controlling diffusion models, making it possible to detect latent biases and perform targeted concept analysis without extensive manual intervention.

Overall, our method provides a scalable and versatile tool for interpreting the intricate latent spaces of diffusion models. By leveraging the power of language-guided exploration, we contribute to the ongoing efforts to make generative models more transparent, interpretable, and ethically responsible. Future work will explore reducing the effect of different random seed initialization on the sampled results, as well as extending the analysis using quantifiable metrics on a broader range of domains.

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
